# Review of Current Prospects for Using *Miscanthus*-Based Polymers

**DOI:** 10.3390/polym15143097

**Published:** 2023-07-20

**Authors:** Nadezhda A. Shavyrkina, Vera V. Budaeva, Ekaterina A. Skiba, Yulia A. Gismatulina, Gennady V. Sakovich

**Affiliations:** 1Laboratory of Bioconversion, Institute for Problems of Chemical and Energetic Technologies, Siberian Branch of the Russian Academy of Sciences (IPCET SB RAS), Biysk 659322, Russia; eas08988@mail.ru (E.A.S.); julja.gismatulina@rambler.ru (Y.A.G.);; 2Department of Biotechnology, Biysk Technological Institute, Polzunov Altai State Technical University, Biysk 659305, Russia

**Keywords:** *Miscanthus*, carbon footprint, cellulose, industrial processing, renewable energy sources, enzymatic hydrolysis

## Abstract

Carbon neutrality is a requisite for industrial development in modern times. In this paper, we review information on possible applications of polymers from the energy crop *Miscanthus* in the global industries, and we highlight the life cycle aspects of *Miscanthus* in detail. We discuss the benefits of *Miscanthus* cultivation on unoccupied marginal lands as well as the rationale for the capabilities of *Miscanthus* regarding both soil carbon storage and soil remediation. We also discuss key trends in the processing of *Miscanthus* biopolymers for applications such as a fuel resources, as part of composite materials, and as feedstock for fractionation in order to extract cellulose, lignin, and other valuable chemicals (hydroxymethylfurfural, furfural, phenols) for the subsequent chemical synthesis of a variety of products. The potentialities of the biotechnological transformation of the *Miscanthus* biomass into carbohydrate nutrient media and then into the final products of microbiological synthesis are also examined herein.

## 1. Introduction

The efficient transition from a fossil fuel-based production system to a renewables-based system will occur chiefly through the use of easily renewable cellulosic raw materials. Plant biomass, as a renewable carbon source, is capable of substituting fossil-based fuels, chemicals, and materials [1]. In recent years, indeed, biomass has been touted not only as an alternative energy source but also as a sustainable platform for the manufacture of reagents and polymers [2,3].

The most essential advantage of using biomass in production processes is its environmental friendliness. Biomass binds carbon dioxide and secures a negative carbon balance as it grows, and the carbon neutrality criteria is achieved when biomass is technologically processed, which is critically important today given the climate crisis [4,5,6].

The most efficient source of biomass is energy crops, and *Miscanthus* has been globally recognized as one of the most significant among them [7,8,9,10]. *Miscanthus* is basically utilized for the production of heat and electric power, and it is actively displacing wood in papermaking and natural cellulose isolation technologies [11,12,13]. However, the range of potential applications of *Miscanthus* is much broader.

The present review is a compilation of the current information on positive experiences of utilizing *Miscanthus* in various human activity fields over the last 15 years.

## 2. Biomass as a Tool towards Carbon Neutrality

The progression of science and technology has appreciably elevated the comfort level of human existence on the planet, and it continues to raise that level. The price we pay for this is the ever-increasing level of energy consumption that is required in order to support basic and secondary human needs. That being said, the inadequacy of carbon-based energy from non-renewable resources (oil, coal, gas) is becoming more and more evident. The dilemma here is that humans are creating more and more energy-consuming devices that make the life easier while, at the same time, traditional energy resources are rapidly running out. Moreover, climate change due to the accumulation of greenhouse gases released by man is becoming more critical [14]. All spheres of human activity are tied to energy consumption, whether they are the manufacture of goods, the maintenance of comfortable living conditions, or the entertainment industry. The pathway to energy problem solving is obvious, and it is globally recognized: it is necessary, in short, to search for renewable energy resources [15], and the more options that are found, the more successful and stable the shift away from our conventional fuels to alternative fuel forms will be.

At present, the most substantial claim against conventional energy sources is that they leave a significant carbon footprint when combusted. Carbon footprint refers to man-made greenhouse gas emissions, basically CO_2_, which are the major cause of excessive global climate change today, especially as the amount of electrical and electronic equipment that is wasted has abruptly risen. Non-renewable energy sources were formed millions of years ago, and they do not provide for the fixation of carbon dioxide but, rather, only its release [16]. The research on this subject notes that if carbon emissions increase at the current rates, humankind will face a 3.2–5.4 °C temperature rise by the end of the century, which may lead to irreversible and destructive ecological consequences, such as deglaciation and the extinction of living species [17].

What promising sources of energy can be considered in the context of overcoming the energy crisis (Figure 1)? Since the humankind has almost run out of the energy provided by the fire element (i.e., non-renewable energy sources that the human being takes out of the earth and burns), we must turn our minds to the other three elements—water, air (wind), and earth (plant-based biomass) [18]. Humankind has been utilizing the energy of water for a long time by creating hydroelectric power plants, and the same is also true of the energy of air, which is utilized via the installation of massive networks of wind turbines. Another energy resource is solar radiation, which human beings have tried to exploit by designing solar batteries. These energy sources are excellent because no human efforts are required to create them as Nature alone makes them. Another issue that is worthy of thought is which of the alternative energy resources is easier to manage? In other words, which energy resource is the easiest to store up, and which resource has a volume that is simpler to predict? The answer is that among the abovementioned alternative energy pathways, biomass conversion is the most predictable and sustainable carbon resource that has the best chance of being a substitute for fossil fuels [19]. The main advantages of biomass as an alternative renewable energy resource are its annual reproducibility in large amounts, its low cost, and its moderate impurities.

What requirements should a plant meet for its biomass to be viewed as the alternative feedstock for the abatement of our carbon footprint? The answer is that the plant must grow fast, produce a high biomass yield for several consecutive years, fixate as much carbon dioxide as possible during photosynthesis, release a lot of energy when it is combusted, and serve as a starting feedstock for chemical and biotechnological transformations into high-value products. Among the plants that satisfy these requirements is the perennial plant *Miscanthus*. *Miscanthus* is characterized by the solar energy conversion type called C4 photosynthesis, which has a higher fixation of CO_2_ compared to C3 plants [20,21]—that is to say, as *Miscanthus* grows, it absorbs more carbon dioxide from the environment, thereby considerably reducing the CO_2_ level in the atmosphere.

## 3. Benefits of Energy Crop *Miscanthus* as a Tool towards Carbon Neutrality

The *Miscanthus* yield capacity attains a 40 t/ha cultivation area and exhibits a high energy release (140–560 Gj/ha) compared to other raw materials [22,23]. In a more severe Russian climate, the biomass yield is also rather high: about 10–16 t/ha a year [24,25]. *Miscanthus* can perfectly grow on marginal or unused lands. In doing so, *Miscanthus* has no tendency of overgrowing in uncontrolled fashion across the entire available territory, and it will thus not supersede the other plants that are customary in that location or disturb the biotic communities [26,27,28,29]. Moreover, *Miscanthus* successfully performs functions that improve the ecology and the environment: it protects landscapes from erosion; it favors organic matter storage in soil, considerably reducing CO_2_ emission [30]; it contributes to phytoremediation of contaminated soils [31]; and, finally, it supports both an active degradation of aliphatic compounds in oil-polluted soils and an increase in bacterial diversity [32]. As it grows, *Miscanthus* enriches the soil with organic substances and enhances the soil respiration. Thus, *Miscanthus* cultivation allows the sequestration of a significant quantity of soil carbon. It has been discovered that *Miscanthus* cultivation can be carbon-negative (i.e., the accumulation of more carbon aboveground and underground than is emitted into the atmosphere) under certain conditions [4,5,6]. This can be particularly relevant to marginal lands where soils are low in carbon, and, hence, the carbon sequestration probability is high. The agronomic, energetic, and ecological efficiencies of *Miscanthus* cultivation are high, then, and there are experimental data on a humus increment in the top soil during *Miscanthus* cultivation and on a higher ratio of energy that is contained in the aboveground biomass when compared to the overall inputs of technical energy for cultivation and harvesting [33,34,35]. The benefits of the energy crop *Miscanthus* as a tool to attain carbon neutrality are listed in Table 1.

The essential merit of *Miscanthus* is its high cellulose content ranging from 36 to 55%, according to different data [12,38,39,40,41,42,43,44,45,46,47]. Moreover, the *Miscanthus* biomass contains valuable constituents, such as hemicelluloses and lignin (Figure 2). Therefore, *Miscanthus* is viewed as the most valuable feedstock for the manufacture of a wide spectrum of reagents and polymers [2,3,45].

According to some forecasts, the utilization of the *Miscanthus* biomass for energy needs will result in a carbon footprint reduction to 30.6 tons CO_2_-eq./ha a year in Central Europe, while this positive trend will constitute about 19 tons CO_2_-eq./ha a year in countries with a cold climate (including Russia) [48]. As per the reported estimate [36], *Miscanthus* cultivation would compensate for greenhouse gas emissions as high as 4.08 tons CO_2_-eq./ha, affecting the global environmental situation favorably.

*Miscanthus* would help reach the carbon-neutral bioeconomy. The core applications of *Miscanthus* are depicted in Figure 3 and detailed in Section 4, Section 5, Section 6, Section 7 and Section 8.

## 4. *Miscanthus* in Energy Production

The simplest energetic application of the *Miscanthus* biomass is the manufacture of fuel pellets. The studies [37,38,39] show that the *Miscanthus* pellets have a lower environmental impact compared to the wood ones, mainly because of less energy consumption during granulation: 1 ton of the *Miscanthus* pellets produce a carbon footprint of 121.6 kg CO_2_-eq., which is about 8% lower than that of the wood pellets [49,50,51]. In order to enhance the heating value, transgenic *Miscanthus* lines with enhanced lignin content were derived [52].

Perić et al. [53] carefully evaluated the pyrolysis process of *Miscanthus* from the viewpoint of ecology and performed an in-depth analysis of three well-to-pump pathways for the production of pyrolytic diesel: a conventional diesel production pathway, a distributed–external pathway in which the diesel stabilization and upgrading processes are separated and hydrogen is generated from natural gas, and an integrated internal pathway in which stabilization and upgrading are combined and hydrogen is generated from the pyrolytic oil fraction. The conventional diesel production pathway had the highest resource consumption, and the distributed-H_2_ external pathway had the highest pollutant emissions. The integrated-H_2_ internal pyrolytic pathway had the lowest specific environmental impact per consumed power unit, but the yield of the resultant diesel in this case was 38% lower than that in the distributed external pathway.

Apart from energy production, *Miscanthus*-derived biocoal can also be utilized for other purposes. Bartocci et al. [54] described a process for biocoal from *Miscanthus* by slow pyrolysis followed by granulation. The estimations showed that the total carbon footprint dropped by 737 kg CO_2_-eq./t dried feedstock. It was also indicated that the obtained biocoal, when used as a soil additive, would be able to greatly improve the soil characteristics: improvements in soil health and fertility, structure, nutrient availability, and water-holding ability, and such a treatment favored a prolonged preservation of soil carbon. Soil carbon sequestration can be viewed not only as a strategy for global climate change mitigation but also as a source of corporate profits through carbon emission trading.

There are research developments that afford bio-oil from the *Miscanthus* biomass by ablative fast pyrolysis [55], and the bio-oil can also be used as a biofuel alone.

Another promising direction in the transformation of *Miscanthus* into biofuel is the pretreatment of *Miscanthus* biomass followed by methane fermentation in order to generate biogas [44,56,57,58]. Dębowski et al. [59] reported a study in which microalgae biomass and *Miscanthus giganteus* were co-fermented. The *Miscanthus giganteus* silage combined with the microbial biomass was found to provide a better C/N ratio than either of the substrates used separately.

Furthermore, because *Miscanthus* biomass has a significant content of cellulose (about 50%), it can be hydrolyzed to monosugars after pretreatment, and, afterwards, alcoholic fermentation can be employed to obtain bioethanol, which can be used as an energy carrier [60] or a platform for further transformation into other industrially valuable components [61].

For the biomass-to-fuel transformation, Agostini et al. [62] used the top of crops that were purposely planted across the banks of basins bordering the croplands to prevent eutrophication. It was noted that the cultivation of *Miscanthus* and willow tree on the buffer strips could remove nutrients from the environment and fixate atmospheric carbon by creating an additional ground absorber. Bioethanol derived from that biomass also had less environmental impact compared to the fossil gasoline that was used [62].

## 5. *Miscanthus* in Construction

An interesting and promising approach to energy efficiency problematics is the development of lightweight and durable composite materials, such as thermal insulation for buildings and constructions, due to the considerable amount of power that is consumed to heat them. It is known that roughly 45% of the global greenhouse gas emissions are caused by building construction and operation [63]. The thermal insulation of buildings under the current conditions of climate change is a well-known strategy for enhancing energy efficiency in buildings. The development of a renewable thermal insulation material can overcome the drawbacks of commonly used insulation systems based on polystyrene or mineral wool. Witzleben [63] evaluated the stability and the thermal conductivity of new insulation materials composed of *Miscanthus giganteus* fibers, a foaming agent, and an alkali-activated binding agent. The data published over the recent years show that the carbon footprint quantity widely ranges from 300 to 3300 kg CO_2_-eq./t. The total carbon footprint of the isolation system based on *Miscanthus* fibers, having properties compliant with the current thermal insulation standards, comes up to 95% of the CO_2_ emissions saving compared to the common systems. Ntimugura et al. [64] proposed the use of *Miscanthus* straw in the manufacture of lightweight concrete blocks for use in wall structures. Another interesting development is the preparation of a self-growing, building insulation, biocomposite material based on *Miscanthus* × *giganteus* and mycelium [65]. The 0.3:1:0.1 composite was composed of *Ganoderma Resinaceum* mycelium, *Miscanthus* × *giganteus* fibers, and potato starch exhibited the best properties. The obtained new composite was found to have comparatively better qualities than the conventional isolation materials—more specifically, a mean density of 122 kg/m^3^, which characterizes the composite as being a lightweight porous material; a high thermal conductivity up to 0.104 W m^−1^ K^−1^, which indicates a high insulating ability; and a significant fire resistance, referring to fire rating EI15 as per the EN13501-2:2003 standard. The new composite thus meets most indoor use requirements.

Lemaire et al. [66] reported the study results for properties of a biocomposite material based on *Miscanthus* × *giganteus* fibers and microbial polyhydroxyalkanoates, fabricated by extrusion and pressure casting. It was pointed out that the addition of reinforcement to the polymeric matrix resulted in composites with higher elastic moduli on the one hand, and a lower tensile strength, on the other hand.

To improve the acoustic performance, Ntimugura et al. [64] prepared composites comprising *Miscanthus* × *giganteus* fibers and lime. Emphasis was made on the particle size of the composites, and it was established that small-size particles afforded *Miscanthus*-lime composites with higher sound absorption factors.

Wu et al. [67] reported the study results of making biocomposite plastics based on *Miscanthus* fibers and oat hulls. The green composites were found to exhibit a relatively high impact ductility and resistance of the melt resulting from the formation of a nanostructured matrix of ultra-high strength.

## 6. *Miscanthus* in Pulp and Paper Industry

*Miscanthus* refers to cellulosic raw material resources, and its biomass contains about 50% cellulose [7,45,46,47]. Cellulose is the most valuable component of *Miscanthus*, and a global trend towards replacing wood cellulose by *Miscanthus* cellulose in the manufacture of cupboards, disposable tableware, and paper is currently being observed worldwide. That being said, it is evident that wood technology cannot be carried over to *Miscanthus*. Over the last 50 years, only ten non-woody plants, including *Miscanthus*, have demonstrated good results from an evaluation of the feasibility of producing cellulose for making paper products [68,69]. Two countries in the world, China and India, have been using about 70% of non-woody plants for paper production since 1990, with *Miscanthus* taking the lead among the plants [70]. Therefore, processes for cellulose isolation from *Miscanthus* biomass are being extensively devised. Tu et al. [71] proposed the pulping of *Miscanthus* × *giganteus* biomass by using a protonic ionic liquid of triethylammonium hydrogen sulfate and 20% water added as the co-solvent. Conditions were found that remove hemicelluloses and lignin, and the pulp contained up to 82% cellulose, which had a high degree of crystallinity (73%) but a lowered degree of polymerization (DP) (a number-average DP of 257).

Tsalagkas et al. [70] reported that the preparation of cellulose pulps from *Miscanthus* × *giganteus* stalks by using hydrodynamic cavitation following an alkaline pretreatment method was able to lower the lignin content by 41.5% and raise the α-cellulose content by 13.87%. The hydrodynamic cavitation was noted to preserve the cellulose fiber length but to concurrently increase the amount of intertwined and twisted fibers.

Barbash et al. [10] obtained *Miscanthus* cellulose by an eco-benign organosolv process in which *Miscanthus* was pulped in a peracetic acid solution as the first step and then alkali-treated as the second step. Nanocellulose (particle size of 10 to 20 nm) was then derived from the resultant material by hydrolysis in a sulfuric acid solution followed by ultrasonic treatment. That study also pointed out that the use of nanocellulose had a positive effect on the physicomechanical properties of paper.

Danielewicz et al. [12] obtained pulps from *Miscanthus* × *giganteus* biomass by the two techniques, soda and kraft pulping. It was eventually found that both the processes yielded cellulose similar in properties to unbleached hardwood kraft pulps (birch, poplar, and hornbeam), and this will make it possible to replace the latter in the manufacture of packaging paper. “The tear strength and Gurley air-resistance of paper handsheets made from *Miscanthus* soda and kraft pulps at freeness of 35–50° SR were good and comparable with those of birch kraft pulp, respectively. All of the *Miscanthus* kraft pulps and the hard and regular *Miscanthus* soda pulps, due to the similarity of their properties to the properties of hardwood kraft pulps (birch, poplar, hornbeam) could probably replace the latter pulps in the production of packaging papers (e.g., sack paper).” Tsalagkas et al. [70] reported that the intermixing of long-fiber (softwood pulp) with short-fiber cellulose (*Miscanthus* pulp) improved paper properties, as the fine particle populated voids to form smoother paper sheets, which is good for printing.

Tsalagkas et al. [70] also reported that delignification combined with ultrasonic treatment allowed for a 41.5% decline in the lignin content and a 13.87% increase in the α-cellulose content. The resultant pulp was fit for papermaking—the average *Miscanthus* fiber length was relatively short (0.45 (±0.28) mm), while the slenderness ratio, the flexibility coefficient, and Runkel ratio values were 28.13, 38.16, and 1.62, respectively. The estimated physical properties of *Miscanthus* pulp handsheets were 24.88 (±3.09) N m g^−1^ as the tensile index, 0.92 (±0.06) kPa m^2^ g^−1^ as the burst index, and 4.0 (±0.37) mN m^2^ g^−1^ as the tear index. Thus, *Miscanthus* can successfully replace hardwood.

There is a process for producing long-fiber cellulose from *Miscanthus* by the hydrotropic method using a saturated solution of sodium benzoate as the solvent of lignin. This method is interesting because it can isolate both cellulose and lignin at the same time [72,73,74]. The hydrotropic method of pulp production involves cooking plant raw materials in neutral concentrated aqueous solutions of sodium salts of various (benzoic, naphthoic, benzenesulfonic, and naphthalenesulfonic) acids, their homologs and derivatives, sodium thiophene carboxylates, hydroaromatic derivatives (e.g., sodium naphtha enates and abietic acids), and salts of various aliphatic aromatic and aliphatic acids. All hydrotropes have a salt-forming effect, and the solubilizing ability of the hydrotropic solution increases with an increase in the salt concentration. The use of water soluble and safe hydrotropic agents meets the “green chemistry” principles. Hydrotropic processing under optimized conditions (180 °C, 5 h) produces pulps in a 42.3% yield, and the lignin, pentosan, and ash contents are 6.1, 6.4, and 3.0%, respectively.

The peculiar feature of *Miscanthus* is its enhanced lignin content, which must be taken into account in its conversion technology [75]. Very attractive results were reported by Singh et al. [76], who devised a method by which paper sheets derived from *Miscanthus × giganteus* cellulose fibers were prepared for potential contact with foods. Paper was hydrophobized with modified lignin, which was also isolated from *Miscanthus × giganteus* biomass via hydroxyethylation with ethylene carbonate, followed by esterification with propionic acid. The results from that study showed that the *Miscanthus* paper coated with esterified lignin holds promise as a hydrophobic food packaging material that can be an alternative to conventional thermoplastics based on fossil fuel.

## 7. *Miscanthus* in Chemical Industry

The essential application of cellulose is tailoring its functionality. The main applications are depicted in Figure 4 [77,78,79,80].

*Miscanthus* can also be involved in various projects as part of the biomass refining concept to produce valuable chemicals, such as 5-hydroxymethyl furfural and furfural, as well as chemicals from mixed phenols [81,82] if emphasis is placed on *Miscanthus* as the source of lignocellulosic biomass. Hydroxymethyl furfural is a promising platform molecule as a substitute for toxic formaldehyde, and it can serve as a feedstock for fungicides or be a part of electron-transfer catalysts [83]. There are data that fuel and food additives can be made with hydroxymethyl furfural as the platform. 5-Hydroxymethylfurfural (5-HMF) is bio-based and can produce polymers with a low refractory deformation temperature, which is a clear advantage for biopolymers. “5-HMF conversion to monomers and polymers can be categorized into three main groups: (1) polymers containing a furan ring: e.g., furan-2,5-dicarboxylic acid (FDCA), 2,5-dihydroxymethylfuran (DHMF), or 5-hydroxymethylfuran-2-carboxylic acid (HMFCA); (2) polymers containing C6 carbon chains with adipic acid or 1,6 hexanediol as building blocks; (3) other polymers made from FDCA with ethylene via the Diels-Alder-reaction e.g., promising building blocks levulinic acid and terephthalic acid” [84]. Hydroxymethyl furfural is also a building block in the manufacture of polyesters, polyamides, and other plastics [85].

Furfural is another bio-based chemical and an essential furan derivative for biochemistry [86]. It is produced from hemicellulose, and it can be a substitute for formaldehyde as well. Among the core applications of furfural is the hydrogenation to furfuryl alcohol. The annual output of furfural ranges from 300 to 700 Kt [87]. Other promising furfural derivatives have also been synthesized in recent years. For instance, a bio-based polyethylene-furanoate (PEF) was obtained by bioconversion of furfural and is able to generate fibers and films, and it can be used for the fabrication of a disposable biodegradable package. 5-Hydroxymethylfurfural derivatives are potential building blocks for step-growth polymers. The aromatic nature of the furan ring gives access to conjugated polymers, especially for optoelectronic applications [86,88]. The phenol mixture that basically contains phenol, catechol, eugenol, and so forth from the lignin fraction also has significant potential, especially in the pharmaceutical industry, for the synthesis of drugs, the manufacture of insecticides, and so on [89].

There are data on the development of a modular concept of biorefinery for the production of hydroxymethyl furfural (HMF), furfural, and phenols from the perennial lignocellulosic *Miscanthus*, which is cultivatable on marginal and dehydrated territories. As per the techno-economic estimations by Götz et al. [90], “regional biorefineries could already offer platform chemicals at prices of 2.21–2.90 EUR/kg HMF at the current stage of development. This corresponds to three to four times the price of today’s comparative fossil base chemicals.” From the standpoint of these authors, such a concept is a competitive option of biorefinery for the conversion of *Miscanthus* into valuable chemicals.

## 8. Biotechnological Conversion of *Miscanthus* Biomass

The greatest opportunities are offered by enzymatic hydrolysis of pretreated *Miscanthus* biomass to derive sugar-containing nutrient media [23,71,81] for the purpose of subsequent microbial transformation into high-value products—for example, bacterial cellulose [91,92,93] or for the extraction of various enzymes from the culture medium [94]. In particular, Xiang et al. [94] developed a process for cellulase production from *Trichoderma reesei* RUT C30 induced by continuous feeding of steam-exploded *Miscanthus* biomass. The experimental results showed that the continuous feeding of lignocellulosic inducers resulted in a higher production of cellulase compared to the fed-batch and intermittent fed-batch strategies. When doing this, the phenomenon of enzyme adsorption onto the substrate surface should be kept in mind [95].

Furthermore, there are studies that allow the evaluation of prospects for using not only *Miscanthus* enzymatic hydrolysis products but also the spent solutions that result from the biomass pretreatment—after *Miscanthus* is pretreated with dilute nitric acid, the spent solution neutralized with ammonium hydrate is employed as a compound lignohumic fertilizer. The growth-regulating activity of the lignohumic fertilizer was studied with pea seeds: it was found that in the dilution ratio ranging from 1:100 to 1:10,000, the germinating power and ability were increased by 2–6% compared to the control, and the root growth increased by 21–29% (that is, there was an auxin-like growth-stimulating activity) [96].

## 9. Conclusions

Given the current trends towards the low-carbon economy as well as the studies on finding alternative pathways to enable carbon footprint reduction, the biomass of energy crops, particularly *Miscanthus* biomass, can be viewed as one of the promising options to come closer to net zero emissions. Considering this crop from the perspective of life cycle assessment, one can certainly predict a reduction in the carbon footprint of humankind if the *Miscanthus* biomass is utilized in any form in view of the following points:(1)*Miscanthus* cultivation alone has a significant ecological effect: the cultivation requires no croplands, so marginal territories or buffer strips can be exploited. By doing so, *Miscanthus* requires no special agrotechnical measures, provides a high yield for several consecutive years, and contributes to a reduction in CO_2_ emission into the atmosphere because of its intensified C4 photosynthesis. Concurrently, the problems of soil remediation against possible contaminations with heavy metals or oil spills are handled.(2)Realizing the potential of *Miscanthus* in different forms in power engineering—from combustion as fuel pellets and biocoal to biotransformation into biogas and bioethanol—would considerably reduce the carbon footprint of mankind.(3)The thermal insulation properties of *Miscanthus* biomass make it possible to use it as a part of building composites, which will cut down energy inputs for the heating of buildings and construction projects.(4)The high cellulose content makes *Miscanthus* biomass a promising feedstock for cellulose extraction and further functionalization.(5)*Miscanthus* biomass can be used as the starting feedstock for fractionation in order to produce high-value chemicals, such as hydroxymethyl furfural, furfural, and phenols.(6)Hydrolysis of the cellulosic constituent of *Miscanthus* biomass can afford carbohydrate nutrient media for the biosynthesis of enzymes of all kinds, for bacterial nanocellulose, for bioethanol, etc.

To sum up, *Miscanthus* cultivation and *Miscanthus* biomass biorefining fully meet the principles of the carbon footprint reduction concept, and they will aid the environmental improvement process on the planet if the developed technologies are adopted.

## Figures and Tables

**Figure 1 polymers-15-03097-f001:**
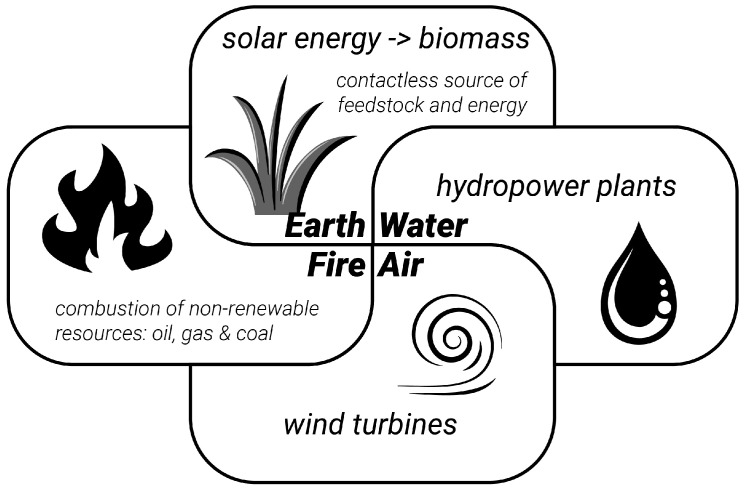
Energy sources to overcome the global energy crisis.

**Figure 2 polymers-15-03097-f002:**
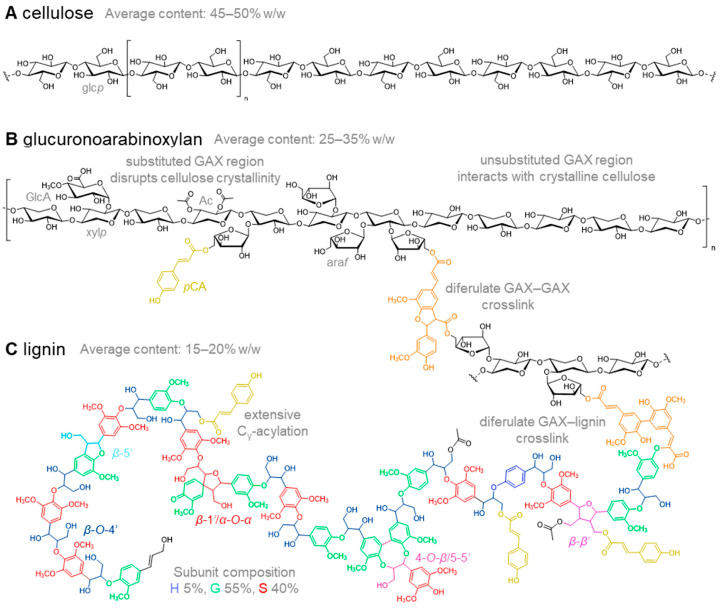
Schematic representation of the major constituents of the *Miscanthus* secondary cell wall: cellulose (**A**), glucuronoarabinoxylan (GAX) (**B**), and lignin (**C**) (reproduced with permission from [45], MDPI, 2021).

**Figure 3 polymers-15-03097-f003:**
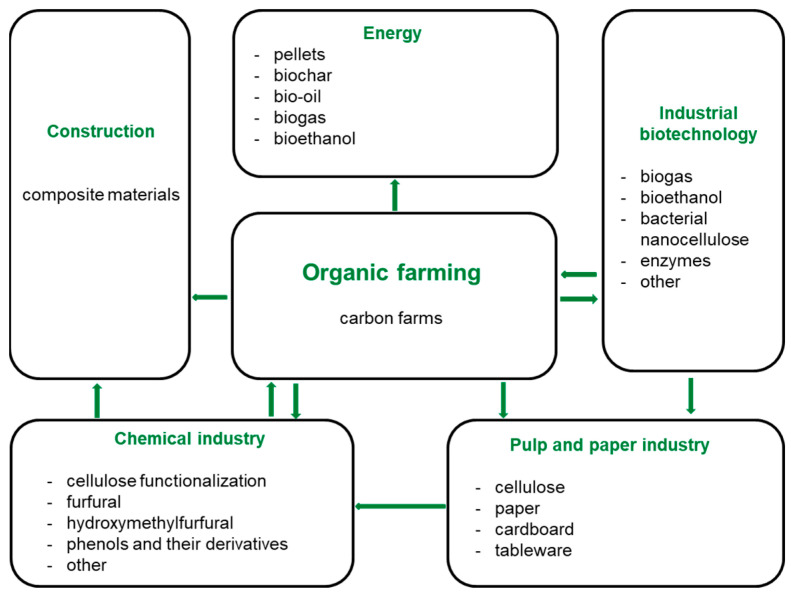
*Miscanthus* in the global carbon-neutral bioeconomy.

**Figure 4 polymers-15-03097-f004:**
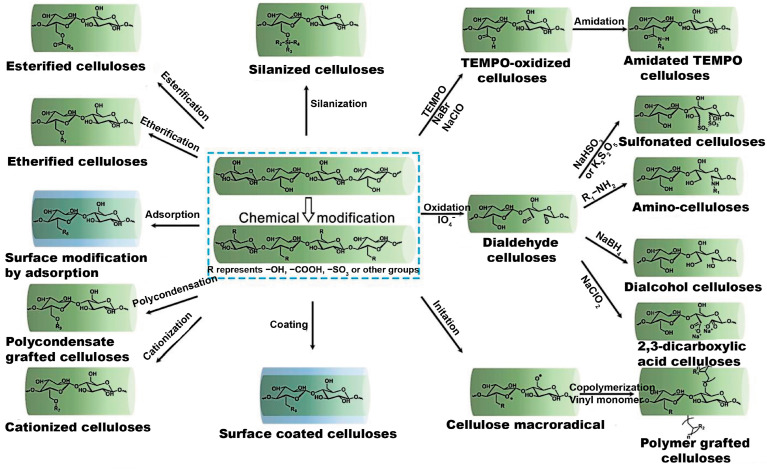
An overview of tailoring the functionality of cellulose (reproduced with permission from [77], MDPI, 2023).

**Table 1 polymers-15-03097-t001:** Benefits of energy crop *Miscanthus* as a tool towards carbon neutrality.

Benefits	Description	Ref.
High yield capacity	Up to 40 tons dry biomass a hectare per annum	[22,23]
Perennial crop	Single planting Plantation operates for 20–25 years	[12,36,37]
Simple agricultural practices	It grows on marginal lands It requires no special agronomical practices It remediates soils It is a non-invasive crop	[26,27,28,29]
C4 photosynthesis	It fixates much of CO_2_ as it grows when compared to C3 photosynthesis	[20,21,30]
Chemical composition	50% cellulose 20% lignin 20% hemicelluloses	[12,38,39,40,41,42,43,44,45,46,47]
Carbon-negative crop	It fixates more CO_2_ during photosynthesis than it releases when converted It builds up the same biomass volume in the underground part as in the aboveground part	[30]

## Data Availability

Not applicable.

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
