# Peer review of "Review of Current Prospects for Using Miscanthus-Based Polymers"

_polymers, 2023, doi:10.3390/polym15143097_

Round 1
Reviewer 1 Report
Dear authors,
The review article is on an interesting topic but it lacks scientific depth in several occasions.
The word count is just above the limit for a Review paper and the first two chapters, which do not directly address Miscanthus, are making up already 30% of the word count. Similarly, the references in these first two chapters make up more than 30% of all references cited in this review.
In my opinion, these sections need to be shortened and the focus needs to be put much more on Miscanthus-based polymers, as the title of the review says.
There is one major flaw in the usage of Figure 2. The figure is absolutely idential to the figure in the cited reference. This is not acceptable, if the citations is not changed to "reprinted with permission from ..." or if the figure is really adapted/changed.
Please be careful with certain statements:
line 26: "Plant biomass is the only renewable source that contains carbon atoms..." what's about fungi? what's about fungi? whats about animals? Please revise!
In the following chapters, there are missing details of the cited literature references in many occasions. The mere mentioning of the references without detailed explaination of the research results in one ore two sentences is not enough for a review paper. Some examples:
line 197: Reference [71] is not in the scope of energy production, (chapter title!) please correct this. The used biochar is not for burning purposes.
line 205: please expand, this statement as such doesnot add any information to the review
line 246 and 247:
"comparatively better qualities" ... what qualities?
"most of the requirements" ... which requirements exactly? this is important information which support the use of Miscanthus in this respective application
line 266-268: is there proof for this statement, like increased numbers of publications, increase in filed patents? please add information
line 270-272: what was the outcome of this pulping method? please add info
line 286: which properties? please add
line 289: which paper qualities? please add
line 290-293: please explain better, there are three references cited.
line 316: please add more information regarding reference [101]
line 326: please add information regarding reference [105], this statement is too vague!
In summary, the manuscript needs to be carefully revised before it could be published.
Further, the English language should be carefully revised and certain wordings should be corrected, a native speaker might be a necessary support Some examples:
line 54, 62, 75, 76, 99, 102: avoid the article the
line 59: "the more options..., the more successful ..."
line 116, 130: uptake is not a verb!
line 241: be is wrong
line 244: consisted is wrong
line 261: the use of strength is wrong
line 323: "It is being produced ..." please revise
line 342: the use of "... such as, for example ..." is wrong
line 357: root growth was stimulated and increased by 21-295? please revise the English language
Author Response
The authors' response to Reviewer 1' comments has been uploaded as a Pdf.

Reviewer 2 Report
Specific comments to the Authors
1. The review gives a good account of the studies related to the current prospects for using Miscanthus-based polymers and is reasonably well prepared. Nevertheless, certain unnecessary statements which are not required in a scientific review are included. These could be deleted as more scientific approach is advisable for such a review.
2. Page 2, lines 54-57: Such sentences could be avoided form a scientific review.
3.page 2, lines 61-68: Recast these lines in a more scientific way and repetitions could be avoided.
4.Pages 2 and 3: The highlighted lines from 78 on p. 2 to 97 on p. 3 could be avoided. These do not have a direct bearing on the scientific content of the review.
5. Page 4, lines116-121: Known scientific facts could be deleted from the review as these are not related to the Miscanthus species.
6.Page 8, line 230: please give a citation for the statement .the percentage seems to be on the higher side.
7. Page 13: Reference Section: Revise this section after deleting the references from the unwanted portions in the text. Also the total number could be reduced to 70-80.
8. Other minor comments included in the highlighted version of the PDF of the review shall be incorporated.

Author Response
The authors' response to Reviewer 2' comments has been uploaded as a Pdf.

Round 2
Reviewer 1 Report
Dear authors,
the revison of the article was done in a satisfying way. All comments were addressed. The review provides a sound reading now and the added information where it was missing before enriched the article.
However, there is an issue with Figure 4, that I didnot realize in the first review round. It was copied as it was pulished in the respective reference. The authors should refer to this Figure in the same way as it was done with Figure 2.
Plus: Both referenced articles are published as open access article under Creative Commons licenses. Please double-check if you need to add the respective licenses of the cited articles.
Best regards and success for the future!
English language is better now and a final English language editing by MDPI, as inicated by the authors, might give the article the satisfying level of English language.
Please rewrite the statement in line 26-27. One suggestion:
Plant biomass as a renewable carbon source is capable to substitute fossil-based fuels, chemicals and materials.
Author Response
The authors' response to Reviewer 1' comments has been uploaded as a Pdf file.
